# Eye Lens Organoids Made Simple: Characterization of a New Three-Dimensional Organoid Model for Lens Development and Pathology

**DOI:** 10.3390/cells12202478

**Published:** 2023-10-18

**Authors:** Matthieu Duot, Roselyne Viel, Justine Viet, Catherine Le Goff-Gaillard, Luc Paillard, Salil A. Lachke, Carole Gautier-Courteille, David Reboutier

**Affiliations:** 1CNRS, UMR 6290, Institut de Génétique et Développement de Rennes (IGDR), Université de Rennes, 35000 Rennes, France; 2Department of Biological Sciences, University of Delaware, Newark, DE 19716, USA; 3CNRS, Inserm UMS Biosit, H2P2 Core Facility, Université de Rennes, 35000 Rennes, France; 4Center for Bioinformatics and Computational Biology, University of Delaware, Newark, DE 19716, USA

**Keywords:** cataract, eye lens, organoid, pathophysiology, *Celf1*

## Abstract

Cataract, the opacification of the lens, is the leading cause of blindness worldwide. Although effective, cataract surgery is costly and can lead to complications. Toward identifying alternate treatments, it is imperative to develop organoid models relevant for lens studies and drug screening. Here, we demonstrate that by culturing mouse lens epithelial cells under defined three-dimensional (3D) culture conditions, it is possible to generate organoids that display optical properties and recapitulate many aspects of lens organization and biology. These organoids can be rapidly produced in large amounts. High-throughput RNA sequencing (RNA-seq) on specific organoid regions isolated via laser capture microdissection (LCM) and immunofluorescence assays demonstrate that these lens organoids display a spatiotemporal expression of key lens genes, e.g., *Jag1*, *Pax6*, *Prox1*, *Hsf4* and *Cryab*. Further, these lens organoids are amenable to the induction of opacities. Finally, the knockdown of a cataract-linked RNA-binding protein encoding gene, *Celf1*, induces opacities in these organoids, indicating their use in rapidly screening for genes that are functionally relevant to lens biology and cataract. In sum, this lens organoid model represents a compelling new tool to advance the understanding of lens biology and pathology and can find future use in the rapid screening of compounds aimed at preventing and/or treating cataracts.

## 1. Introduction

The lens, in conjunction with the cornea, is responsible for the focusing of light onto the retina, thus creating a clear image [1]. It is a fully transparent biological tissue that involves extreme cell differentiation processes. At the histological level, the lens is composed of an anterior monolayered epithelium containing proliferating cells in the equatorial region that later exit the cell cycle and progressively differentiate into fiber cells [2]. These latter cells form the lens cortex once the differentiation process is complete. To achieve lens transparency, fiber cells lengthen extensively (~1000X), produce large amounts of refractive proteins called crystallins, and eliminate their organelles, including their nuclei [3,4]. Lens clouding or cataract is the leading cause of blindness worldwide. While the primary reason for the development of cataracts is aging, they can also be induced by environmental factors or have a congenital origin, often triggered by genetic predispositions or abnormalities [5,6]. To date, the only treatment for cataracts is surgery, which consists of replacing the clouded lens with an artificial implant. Although it is effective, surgery is costly and can have side effects that are far from harmless [7]. Therefore, efforts to develop drugs to treat cataracts have been initiated [8,9,10]. Animal models such as zebrafish [11], Xenopus [12], chicken or mammals, namely, rodents, dogs or macaques [13], are used for the study of lens pathophysiology. However, a major bottleneck toward developing anti-cataract drugs remains the lack of an adequate biological model for intensive drug screening.

In recent years, biology and medicine have undergone a revolution with the advent of particular three-dimensional (3D) cultures called organoids [14]. These are in vitro cellular models that mimic several aspects of the structure and function of the corresponding organ. Lens epithelium explants were a first generation of 3D lens cultures [15]. Later on, lentoid bodies, which are 3D cellular structures emerging from various types of 2D stem cell cultures [16,17,18], or individual micro-lenses grown from lens epithelial cells or pluripotent stem cells [19,20] were described. Although these models have very interesting molecular and/or optical characteristics, they do not exhibit any particular organization reminiscent of the histology of the lens [21]. Moreover, they often require sequential treatments via individual or combined growth factors and remain tricky and time-consuming to implement. Consequently, they generally (except for the paper from Murphy and colleagues [19]) do not allow for high-throughput studies.

The goal of the present study was to develop a mammalian organoid lens model that could be generated rapidly and is more convenient to use. As a starting point, we considered a previous paper, which shows that lens epithelium can regenerate a functional lens after its ablation in several vertebrate models [22]. This capacity relies on the presence of lens stem or progenitor cells that sustain self-renewal. A characterization of these cells demonstrated that they express *Pax6* and *Bmi1* and that they are able to spontaneously generate lentoid bodies. The 21EM15 mouse lens epithelial cell (LEC) line expresses *Pax6* and *Bmi1* and can spontaneously form lentoid bodies in vitro [23,24]. However, these lentoid bodies have not been characterized, and the culture conditions needed to controllably induce 21EM15 cells to become such 3D structures have not been defined.

Therefore, in the present study, we sought to derive the culture conditions that could generate lens organoids from 21EM15 cells *en masse*. Further, we sought to undertake their detailed characterization to evaluate their utility in studying genes and pathways relevant to lens biology and pathology. Our work indicates that by using simple 3D culture conditions, we can generate numerous lens organoids in a short period of time. These organoids show very interesting optical properties and recapitulate lens physiology at the molecular, histological and cellular levels. In addition, our results demonstrate the possibility to induce various types of opacities, thus mimicking cataract, in these organoids. As a whole, the 21EM15 organoids should therefore provide the lens community with a compelling new model to advance the understanding of lens biology and pathology. From a clinical point of view, although derived from mice, these organoids can potentially be used to screen compounds that could have effects on the prevention and/or treatment of cataracts.

## 2. Materials and Methods

### 2.1. Cell Culture

21EM15 (obtained from Dr. John Reddan, Oakland University, Rochester, MI, USA) and HaCat cells (ATCC) were cultured in DMEM with 4.5 g/L glucose, L-glutamine and sodium pyruvate (Life Technologies, Carlsbad, CA, USA) including 10% Fetal Bovine Serum (Eurobio, Les Ulis, France) and 1% penicillin-streptomycin (Life Technologies, Carlsbad, CA, USA). A253 cells (ATCC) were cultured in McCoy 5A with 4.5 g/L glucose, L-glutamine and sodium pyruvate (Life Technologies, Carlsbad, CA, USA) including 10% Fetal Bovine Serum (Eurobio, Les Ulis, France) and 1% penicillin-streptomycin (Life Technologies, Carlsbad, CA, USA). All three cell lines were cultured in 100 mm cell culture-treated Petri dishes (Corning) with 10 mL of culture medium. The cells were grown at 37 °C in a water-saturated atmosphere with 5% CO_2_. These cells grow well in these conditions and are usually 80% confluent after three days in culture (after 10% seeding). Cells were passaged three times per week.

### 2.2. Organoid Culture

Round-bottom 96-well plates were coated with poly(2-hydroxyethylmethacrylate) (Polyhema) (Sigma, Saint-Louis, MO, USA) 1 week before cell culture. Polyhema was first dissolved at 50 mg/mL in 95% EtOH. This stock solution was then diluted to 30 mg/mL with absolute ethanol. For coating, each well, except for the outermost ones, was filled with 50 μL of a 30 mg/mL solution of Polyhema before the plate was allowed to dry overnight. For culture, the outermost wells were filled with 200 μL PBS to avoid evaporation. The remaining wells were seeded with 10,000 cells and filled with 200 μL culture medium for 10 days prior to experiments.

### 2.3. Histology and Image Acquisition

For histological analyses, organoids were washed with 1X PBS, and then fixed for 24 h in 4% pH 7 buffered formalin and processed for paraffin wax embedding in an Excelsior ES automaton (Thermo Scientific, Waltham, MA, USA). Paraffin-embedded tissue was sectioned at 4 µm, mounted on positively charged slides and dried at 58 °C for 60 min. Immunohistochemical staining was performed on the Discovery ULTRA Automated IHC stainer (Roche Diagnostics, Rotkreuz, Switzerland) using the Ventana detection kit (Ventana Medical Systems, Tucson, AZ, USA). For fluorescent labeling, following deparaffination with Discovery wash solution (Ventana Medical Systems, Tucson, AZ, USA) at 75 °C for 8 min, antigen retrieval was performed using Ventana Tris-based buffer solution, pH 8, at 95 °C to 100 °C for 40 min. Endogen peroxidase was blocked with 3% H2O2 for 12 min. After rinsing, slides were incubated at 37 °C for 60 min with primary antibodies. Signal enhancement was performed using a secondary HRP-conjugated antibody at 37 °C for 16 min and DISCOVERY Rhodamine Kit (Roche Diagnostics, Rotkreuz, Switzerland) for 8 min. For fluorescence multiplex labeling, slides were prepared as follows: For the first sequence, following deparaffination with Discovery wash solution (Ventana Medical Systems, Tucson, AZ, USA) at 75 °C for 8 min, antigen retrieval was performed using Ventana proprietary, Tris-based buffer solution, pH 8, at 95 °C to 100 °C for 40 min. Endogen peroxidase was blocked with 3% H_2_O_2_ for 12 min. After rinsing, slides were incubated at 37 °C for 60 min with primary antibody, rabbit anti-PROX1. Signal enhancement was performed using a goat anti-rabbit HRP at 37 °C for 16 min and DISCOVERY Rhodamine Kit (542–568 nm) for 8 min. For the second sequence, slides were neutralized with Discovery inhibitor (Ventana Medical Systems, Tucson, AZ, USA) for 8 min. After rinsing, slides were incubated at 37 °C for 60 min with primary antibody mouse anti-JAG1. Signal enhancement was performed using a goat anti-mouse HRP at 37 °C for 16 min and DISCOVERY cy5 Kit for 8 min. For the third sequence, slides were neutralized with Discovery inhibitor (Ventana Medical Systems, Tucson, AZ, USA) for 8 min. After rinsing, slides were incubated at 37 °C for 60 min with primary antibody, rabbit anti-PAX6. Signal enhancement was performed using a goat anti-rabbit HRP at 37 °C for 16 min and DISCOVERY Fam Kit for 8 min. DAPI staining was used to visualize DNA/nucleus. For chromogenic labeling, following deparaffination with EZ Prep (Roche Diagnostics, Rotkreuz, Switzerland) at 75 °C for 8 min, antigen retrieval was performed using CC1 buffer (Roche Diagnostics, Rotkreuz, Switzerland), pH 8.0, at 95 °C to 100 °C for 40 min. Endogen peroxidase was blocked with 3% H_2_O_2_ for 12 min. After rinsing, slides were incubated at 37 °C for 60 min with primary antibodies. Signal enhancement was performed using a secondary HRP-conjugated antibody at 37 °C for 16 min and revealed using the OmniMap DAB kit (Roche Diagnostics, Rotkreuz, Switzerland). The slides were counterstained with the Mayer’s hematoxylin. Antibodies were as follows: Ki67, ab16667 dilution 1/200 (Abcam, Cambridge, UK); Cleaved Caspase 3, #9661 dilution 1/250 (Cell Signaling Technology, Danvers, MA, USA); Lamin B1, A16909 dilution 1/200 (Abclonal, Woburn, MA, USA); Laminin Z0097 dilution 1/200 (Dako, Glostrup, Denmark); PAX6 AB2237 dilution 1/200 (Sigma, Saint-Louis, MO, USA); JAG1 sc390177 dilution 1/200 (Santa Cruz, Dallas, TX, USA); PROX1 925202 dilution 1/200 (BioLegend, San Diego, CA, USA); CRYAB ADI-SPA-223 dilution 1/200 (Enzo Life Science, Farminfdale, NY, USA); HSF4 HPA048584, dilution 1/200 (Atlas Antibodies, Voltavägen, Sweden); TOMM20 ab186735 dilution 1/5000 (Abcam, Cambridge, UK); Fibrillarin 32639 dilution 1/100 H2P2 (Cell Signaling Technology, Danvers, MA, USA). HES staining was realized on an ST 5020 automaton (Leica, Wetzlar, Germany). Bright field images were acquired using a digital slide scanner Nanozoomer (Hamamatsu, Shizuoka, Japan), while fluorescence microscopy images were acquired using a DeltaVision Elite setup equipped with a Nikon IX71 microscope and a CoolSnap HQ camera (Imsol, Preston, UK).

### 2.4. Transparency and Light Focalization

Transparency was assessed by placing the organoids on an electron microscopy copper grid (mesh 300) for 30 s and imaging them with an AZ100 macroscope (Nikon, Tokyo, Japan). Light focalization was quantified using an Axio Observer inverted microscope (Zeiss, Iena Germany). Briefly, a stack of images starting from the focus and progressively lowering the objective under the sample was acquired. For each image of the z-stack, the maximum light intensity at the center of the spheroid and the mean intensity in the field around the spheroid were quantified. The ratio of the maximum light intensity to the mean intensity was then calculated and plotted to produce a graph.

### 2.5. 21EM15 2D and 3D RNA Isolation

RNAs were isolated using the Nucleospin kit (Macherey-Nagel, Düren, Germany) from a 100 mm Petri dish of 21EM15 cell culture at 80% confluence (2D) or 60 organoids (3D) for each replicate.

### 2.6. Laser Capture Microdissection

For each replicate, 120 organoids were washed with 1X PBS and pelleted before being included in OCT and then snap-frozen using a SnapFrost 80 deep freezer (Excilone, Elancourt, France). The frozen OCT block was then mounted onto a cryostat (Leica, Wetzlar, Germany) and cut to obtain 10 μm sections. The sections were then deposited on polyethylene naphthalate (PEN) membrane frame slides. OCT was removed via multiple washes: 2 washes in 70° ethanol (−20 °C; 5 min), 1 wash in 90% ethanol (RT; 20 min), 1 wash in 100% ethanol (RT; 20 min) and 3 washes in 100% Xylene (RT; 1 min). The internal or external regions of the spheroids were microdissected using an Arcturus XT laser capture microdissection setup (Excilone, Elancourt, France). The RNA was isolated from these samples using the Arcturus PicoPure kit (Excilone, Elancourt, France).

### 2.7. 3′ End RNA Sequencing (RNA-Seq) and Analysis

Libraries were prepared from the extracted RNAs using the QuantSeq 3′ mRNA-Seq library Kit (Lexogen, Vienna, Austria). The 3′ end seq library was sequenced (strand-specific, 150 bp) using Illumina NovaSeq 6000. Quality of the sequence was validated using FastaQC, and only sense reads were used for the analysis. The RNA-seq data are available on the NCBI Gene Expression Omnibus (GEO) database under GSE228547 series. The adaptor sequence and the poly(A) tail were trimmed from the raw sequences, and only read lengths greater than 20 nucleotides were retained. Trimmed sequences were aligned using the STAR software (STAR (v2.7.8a)) [25] onto the mouse genome (GRCm38.p6). Only uniquely mapped reads were conserved for the analysis. Reads were associated with genes using FeatureCounts (v1.6.0) [26]. For differential gene expression analysis, only genes with an expression of >0.2 cpm (counts per million) were considered. The R package, edgeR (v3.32.1) [27], was used to identify significantly differentially expressed genes (DEGs), with the following as as cut-offs: |Fold Change| (FC) > 2 (|logFC| > 1) and a False Discovery Rate (FDR) < 0.05.

### 2.8. Gene Expression Data Analysis

To determine the pattern of expression of the 2D or 3D DEGs in the mice embryonic lens, we used microarray data from the iSyTE 2.0 database [28] to identify genes that exhibited lens-enriched expression in normal lens development across stages E14.5 to P0. As described previously, a comparison of global gene expression data between lens and whole embryonic body tissue (WB) allows for the estimation of lens-enriched expression. To compare the regional transcriptomic profile between organoid and lens, we used previously generated RNA-seq data to identify DEGs with an expression profile specific to isolated FC or LEC [29]. These data correspond to WT mice at the E14.5, E16.5, E18.5 and P0.5 stages. The identification of genes with an expression profile specific to FC or LEC was based on cut-offs of *p*-value adjusted to <0.05 and |FC| > 2.

## 3. Results

### 3.1. 21EM15 Spheroids Are Transparent and Have the Ability to Focus Light

To test their ability to grow under 3D culture conditions, we seeded 21EM15 cells in 96-well culture plate wells coated with polyhema. Twenty-four hours after seeding, the vast majority of the cells were assembled into round spheroids, with no isolated cell being observed. Upon subsequent culture, these spheroids grew in size (Appendix A), acquiring an ovoid asymmetric shape between days 3 and 7. Thereafter, culturing the spheroids beyond day 10 only resulted in limited changes in their overall appearances (Appendix A). At day 10, the 21EM15 spheroids were transparent, contrary to the spheroids generated with two other epithelial cell lines grown in the same conditions, namely, human epithelial keratinocytes (HaCaT cells) or head and neck squamous cell carcinoma A253 cells (Figure 1A). We then tested the capacity of the 21EM15 spheroids to focus light following a previously described approach [19]. Briefly, we imaged the spheroids at different z-positions starting from the focus (Figure 1B). A very bright light spot was observed at the center of the 21EM15 spheroids at a specific z-position (Figure 1C), suggesting that they had acquired properties to focus light. We quantified the light-focusing ability as the ratio of the maximum light intensity at the center of the spheroid to the mean intensity around the spheroid (Figure 1C,D). This ratio reached values well above 1 at z-positions below the focus in the 21EM15 spheroids, confirming their capacity to focus light (Figure 1D); this was not observed in the HaCaT spheroids (Appendix A).

### 3.2. Transcriptome Analysis of 21EM15 Spheroids Reveals Strong Similarities with Lens Development

During their growth, the 21EM15 spheroids acquire an asymmetric shape and optical properties to focus light. To gain insights into the molecular modifications associated with these morphological changes and assess whether they may be relevant to lens development, we compared the transcriptome landscapes of 21EM15 spheroids with those of mouse lenses at several stages of development. We profiled the gene expression in the 21EM15 spheroids via 3′ end RNA sequencing (Appendix A) and we retrieved the gene expression levels of mouse lenses from the iSyTE 2.0 database [28]. In iSyTE 2.0, “lens-enrichment” is estimated as the log-ratio of gene expression in the lens to gene expression in the whole embryonic body (WB). Using a WB comparative analysis, we similarly estimated gene enrichment in 3D 21EM15 cultures. We compared the 10% most enriched genes in 3D cultures (*n* = 1032; 10,320 genes in total) with the 10% most enriched genes in the E14.5 lenses (*n* = 1032). We found that 198 genes were present at the overlap of the two datasets, which was far above what was expected by chance (p = 8.7 × 10^−22^; hypergeometric test). Irrespective of the threshold set to classify the genes as top-enriched (between 0 and 10%), the number of genes observed in the overlap of top-enriched genes in the 3D cultures and top-enriched genes in the E14.5 lenses largely exceeded what was expected (Figure 2A). Nine genes were present in the overlap of the top 1% most enriched genes in the E14.5 lenses and the top 1% most enriched genes in the 3D cultures, whereas only one gene was expected by chance. Among these genes were *Cryab*, *Six3*, *Adamtsl4*, *Cp* (encoding ceruloplasmin), *Crim1*, *Dkk3* and *Nupr1*, all of which are known to be directly linked to lens pathophysiology (Figure 2A). We obtained similar results for all lens development stages present in iSyTE 2.0 (E10.5, E12.5, E14.5, E16.5, E17.5 and E19.5; data not shown). Together, these results reveal an overlap in the expression of key genes between 21EM15 spheroids and normal lenses.

Next, we wanted to assess the contributions of the culture conditions (3D vs. 2D) to gene expression, especially as it relates to the lens. To carry this out, we profiled gene expression in the 21EM15 2D cultures via 3′ end RNA sequencing in the same conditions as the 3D spheroids (Appendix A). A principal component analysis and a hierarchical clustering analysis showed that the 2D and 3D samples cluster separately from each other (Appendix A). We therefore used these datasets to identify differentially expressed genes (DEGs). At FDR = 0.01 and log2 (Fold Change) > 1 in absolute value, this analysis uncovered 291 genes that were elevated and 191 genes that were reduced in the 3D spheroids (Figure 2B). The elevated 291 genes are referred to as “3D genes”, and the reduced 191 genes are referred to as “2D genes”. As expected, the heat map of these 482 DEGs showed a clear separation between these two sets of genes (Figure 2C). Several genes elevated under 2D conditions of growth relate to the cell cycle, such as *Ccnb1*, *Ccnb2*, *Ccnb1*, *Ccne2* and *Cdc20* (Figure 2B). This indicates that culturing 21EM15 in 3D conditions reduces the expression of cell cycle genes comparted to growth under 2D conditions. Conversely, several genes elevated under 3D growth conditions are relevant to lens development (Figure 2B). These include *Apoe*, *Aqp1*, *Cdkn1b*, *Cp*, *Ctsl*, *Cxcr4*, *Lama4*, *Maf*, *Map1lc3a*, *Notch3*, *Psen2*, *Wnt6* and *Wls.*

Finally, we used iSyTE 2.0 to examine the expression of these DEGs in normal lens development [28]. On average, 2D genes are more expressed than 3D genes in early lens developmental stages (e.g., E10.5, when the lens placode has invaginated into a “lens pit”), and their expression decreases as the lens progresses in development (Figure 2D). Conversely, the mean expression of 3D genes increases progressively in normal lens development (Figure 2D). Hence, switching the culture conditions of 21EM15 cells from 2D to 3D spheroids partly recapitulates gene expression changes in normal lens development. Together, these data show that culturing 21EM15 in 3D conditions reinforces their similarity with normal mouse lenses, particularly at later stages of development.

### 3.3. 21EM15 3D Cultured Cells form Multilayered Lens Organoids

The lens is surrounded by a basal lamina called the capsule. Under the capsule and starting from the anterior pole of the lens, there is a quiescent epithelium, containing cells that proliferate in the “germinal zone” and exit the cell cycle at the “transition zone” to initiate differentiation into fiber cells that contribute to the bulk of the lens. Differentiation into fibers cells is characterized by the remodeling of the cytoskeleton, leading to the elongation of the cells, accompanied by high levels of expression of key lens proteins (e.g., crystallins, membrane proteins, etc.) and the progressive loss of cellular organelles [3,4]. As 21EM15 cells that are cultured as 3D spheroids express genes associated with lens differentiation, we wanted to characterize their structures to identify whether different cell types emerge through a process of differentiation. A histological analysis showed that twelve hours after seeding under 3D conditions, the 21EM15 cells clustered to form a non-organized flattened structure (Figure 3A). Twenty-four hours after seeding, the structure was spherical with a rather homogenous cellular content. Ten days after seeding, the spheroids showed a different appearance with distinct zones: an external zone with round nuclei, an intermediate zone with elongated nuclei, and an internal zone with cells characterized by an intense pink cytoplasm and small compacted nuclei (Figure 3A). Importantly, the internal zone was off-centered, revealing that the initial central symmetry of the spheroid was broken during the 10-day culture (Figure 3A). This is in line with previous observations that the spheroid acquires an ovoid shape after a few days of culture (Appendix A).

To evaluate cell compartmentalization in the core of the spheroid, we visualized the cells’ boundaries using WGA (wheat germ agglutinin), a membrane marker. The core of the spheroid is composed of highly compacted cells when compared to the cortex (Figure 3B). This indicates that the cells located at the central region of the spheroid undergo a phenomenon of packing. As controls, histological sections of the HaCaT and A253 spheroids were generated (Figure 3C). After 10 days of culture, the specific organization of cells was not observed in these controls. HaCaT spheroids are made of cells roughly aggregated and cavities likely filled with an extracellular matrix, while A253 spheroids are made of homogenously distributed cells with a clear pink cytoplasm and some spots of necrotic cells (Figure 3C). As 21EM15 spheroids constantly grow over a period of more than 10 days (see Supplemental Appendix A), we wanted to determine which cells were responsible for their growth. Staining with the KI67 antigen, an established marker of proliferation, shows that proliferating cells are essentially localized in the external region of the spheroid and that cells stop proliferating once their nuclei are elongated (Figure 3D). The observation that only a few cells are able to proliferate is consistent with the finding that many cell cycle-related genes are down-regulated in 3D cultures (Figure 2B). Cells undergoing apoptosis were not observed after 10 days of culture (Figure 3E).

The above data indicate that 21EM15 spheroids are made up of at least three distinct regions. We next sought to determine the gene expression landscape within the different layers of the spheroids. Toward this goal, the most internal and external regions were isolated by laser capture microdissection (LCM) and subjected to 3′ end RNA sequencing (Appendix A). Due to technical limitations, we were unable to microdissect the intermediate region. We retained *n* = 3 external region samples and *n* = 4 internal region samples based on PCA (Appendix A). We identified 793 DEGs (FDR = 0.01, and log2 (Fold Change) > 1 in absolute value). Of these, 465 exhibited enriched expression in the external region and 328 exhibited enriched expression in the internal region (Figure 3F). The heat map of these 793 DEGs separates “external genes” from “internal genes” (Supplemental Appendix A). Among the genes that are overexpressed in the internal region, we found genes known to be expressed in fiber cells like *Ank2*, *Atp1b1*, *Cap2*, *Eya1*, *Fundc1*, *Fzd6*, *Hsf4*, *Jag1*, *Maf*, *Meis1*, *Prox1*, *Tdrd7* and *Wls*. Among the genes that are overexpressed in the external region, we found genes known to be expressed in lens epithelial cells like *Ccna2*, *Ccnb1*, *Ccnb2*, *Ccnd1*, *Ccne2*, *Cdc20* and *Cdk1*.

To globally assess the resemblance of the internal and external regions of 21EM15 spheroids with lens fiber and epithelial cells, respectively, we retrieved the transcriptomic data from microdissected E14.5 epithelial cells and lens fiber cells [30]. Of the 793 DEGs, 378 are also enriched either in E14.5 lens epithelium or fiber cells. The “external genes” (negative log_2_(FC) in Figure 3G) are enriched in lens epithelial genes (green spots), whereas “internal genes” (positive log_2_(FC) in Figure 3G) are enriched in lens fiber cells (orange spots). The contingency table shown in Figure 3H confirms this bias (*p* = 2.1 * 10^−6^; chi-squared test). These data confirm that the transcriptome of the internal region resembles that of lens fiber cells and the transcriptome of the external region resembles that of lens epithelial cells. Further, while 2D cultures of 21EM15 cells have overlapping expressions with lens epithelial cells, growing the cells in 3D culture conditions commits the internal cells toward a differentiation program overlapping with lens fiber cells. Taken together, our results show that the 3D 21EM15 cultures self-organize and establish different cell types expressing specific gene sets, thus mimicking certain aspects of lens development. Moreover, these structures are able to break their original central symmetry to establish axial symmetry, which is characteristic of organoid development [31,32,33]. Thus, henceforth, the 3D 21EM15 cultures will be referred to as 21EM15 “organoids”.

### 3.4. Morphological Organization of 21EM15 Organoids Partially Recapitulates Lens Patterning

Different regions or cell types within the lens can be characterized by the expression of specific markers. From the transcriptomic studies described above, we identified a subset of key genes involved in lens development to be differentially expressed between the inner and outer regions (see Appendix A). Our next objective was to confirm that key lens genes have specific expression patterns that are relevant to the organization of a whole lens. For this purpose, we examined the expression and the localization of structural components such as Laminin, a major component of basal lamina including the lens capsule, and αB-Crystallin (CRYAB), a major component of lens fiber cells in later developmental stages [34,35]. We also examined transcription factors such as PAX6, PROX1 (elevated in fiber cells) and signaling molecules such as JAG1 [1,36,37]. Laminin was found to be present in two or three layers of the outermost cells (Figure 4A). It was also present in the most peripheral part of the lens, but only in the form of a single outer basal lamina [35]. Immunostaining showed that nuclear PAX6 is present in an internal region surrounding the central core of the organoid, composed of highly compacted cells (Figure 4B). Consistent with symmetry breaking, it then extends toward the central axis. JAG1 is also found in an asymmetric distribution, as it is enriched in the membranes of cells localized in two lateral areas surrounding the central axis of the organoid (Figure 4B). PROX1 is present in two regions: in the cytoplasm of cells that most strongly express *Jag1*, and in the nuclei of cells that more weakly express *Jag1* and are localized along the central axis (Figure 4B,C). Finally, αB-Crystallin is low/absent in the external cell layers but high in the cortex and the core of the organoid with both cytoplasmic and nuclear localization (Figure 4D).

One of the most striking characteristics of fiber cell differentiation is the progressive degradation of cellular organelles such as nuclei and mitochondria [38]. In the lens, these various events can be highlighted by the observation of components of the nuclear envelope and of the mitochondria or by the expression of specific transcription factors [39,40,41,42]. We sought to examine whether similar cellular changes occurred in 21EM15 organoids. We found that Lamin-B1 (LMNB1), a component of the nuclear envelope, progressively disappears from the exterior to the interior of the organoid (Figure 5A). While Lamin-B1 labeling surrounds the nucleus in a continuous manner in the outermost cells, this labeling becomes more and more discontinuous toward the inner region of the organoid until it completely disappears. Concomitantly, we observed a gradual change in the nuclei shape accompanied by chromatin compaction evoking pyknosis (Figure 5B). These nuclei are not transcriptionally active in cells located in the center of the organoids (Figure 5C), as indicated by the progressive loss of nuclear Fibrillarin (FBL), which is considered as a marker of the transcriptional status of fiber cell nuclei [43]. Mitochondria are also in a process of degradation, as indicated by the decrease in TOMM20 labeling, which is a constituent of the mitochondria external membrane, toward the center of the organoid (Figure 5D). Conversely, the expression of *Hsf4,* a gene encoding transcription factor whose downstream targets are considered to be involved in organelle degradation [40,44], increases in the core region relative to the outer region (Figure 5E,F).

Taken together, these results suggest that, similar to the cellular and morphological changes accompanying lens development, the 21EM15 organoids are organized into specific expression domains for key lens proteins like αB-Crystallin, PAX6, PROX1 and JAG1. Moreover, the cells lying in the internal-most region commit to a process of organelle degradation reminiscent of what is observed in the whole lens. Our results thus show that 21EM15 organoids recapitulate specific molecular aspects and the morphological organization of the lens.

### 3.5. 21EM15 Organoids Model Lens Cataract

As the organoids described above present interesting optical, morphological, histological, molecular and functional characteristics, we explored if they could be utilized as a model to uncover the pathophysiology of the lens. Cataract can be induced by H_2_O_2_ or hypertonic NaCl treatments in dissected lens [45]. Therefore, we incubated 8-day-old 21EM15 organoids for 48 h with these compounds (at previously used concentrations) and evaluated their transparency and light-focusing ability. We found that H_2_O_2_ does not trigger changes in transparency or light-focusing ability for concentrations ranging from 0 to 350 μM, but organoids become opaque and cease to transmit light at concentrations above 500 μM (Figure 6A,B). Increasing concentrations of NaCl (from 1.25% to 1.7%) gradually reduce organoid transparency (Figure 6A), but only the highest concentration has a significant impact on light-focusing ability (Figure 6C). We next sought to examine whether the 21EM15 organoid model could be applied to test the functions of genes associated with cataract. We previously showed that *Celf1* deletion in a germline or lens conditional manner causes early-onset cataract in mice [46]. We therefore tested the transparency and light-focusing ability of organoids made from 21EM15 cells stably expressing an shRNA targeting the *Celf1* gene [46,47]. Interestingly, *Celf1* knockdown reduces organoids’ transparency, as observed in mice deficient in *Celf1* (Figure 6D). It also reduces the light-focusing property of 21EM15 organoids (Figure 6E). All together, these results demonstrate that 21EM15 organoids are new models that can be used to study cataract.

## 4. Discussion

In the present study, we developed a mouse 3D lens model that can be rapidly generated and can be applied to study processes relevant to lens biology. It has specific optical properties, including transparency and light-focusing ability (Figure 1). We characterized this model using a combination of histological, transcriptomic and immunohistochemical/immunofluorescence approaches. This analysis revealed that, similarly to the lens, 21EM15 lens organoids comprise three main regions, including a peripheral layer, an intermediate part and a core region. Transparency and light-focusing ability indicate that even though the histological organization of the 21EM15 organoids is simplified compared to a true lens, it is sufficient to bring relevant optical properties. In addition to transparency and light-focusing ability, a refractive index measurement, as described by Young and colleagues [48], could also be very useful in the future to assess the influence of cellular organization on organoid optics.

When grown in 2D culture conditions, 21EM15 cells express typical LEC genes [23]. In the present study, we used 3′ end RNA-seq to profile the transcriptomes of 21EM15 2D and 3D cultures, and of laser-microdissected internal and external regions of the organoids. These analyses show that, regardless of how they are grown (2D or 3D), 21EM15 cells express various crystallin genes, such as *Cryab*, *Cryba4*, *Crybg1*, *Crybg3*, *Cryz*, *Cryzl2* and *Cryl1* (see Appendix A). Terrell and colleagues [23] showed that various α and βγ crystallin transcripts (*Cryaa*, *Cryab*, *Cryba4* and *Crygs*) are expressed in 21EM15 cells. Using 3′ end RNAseq, we could confirm the expressions of *Cryab* and *Cryba4* but were unable to confirm the presence of *Cryaa* and *Crygs* in 2D or 3D 21EM15 cells. This may be related to the different methods used (Microarray vs. 3′ end RNA-seq) [23]. Nevertheless, both studies show that *Cryba4* is expressed in these cells. These findings, along with the property of CRYBA4 protein to form homodimers, suggest that it likely makes a key contribution toward the refractive properties observed in the organoids. This is further likely because transcripts encoding gamma crystallins (*Cryga*, *Crygb*, *Crygc*, *Crygd* and *Cryge*)—which are the other major contributors for transparency in the developing lens—are not significantly expressed in these cells. Interestingly, this suggests that the lens organoids exhibit a gene expression strategy for transparency (i.e., high *Cryba4*) that is observed in later stages in the lens. Further, while they are expressed, other crystallin encoding genes such as *Crybg1* and *Crybg3* are less likely to contribute to the refractive properties of the organoids based on previous findings [49,50]. Thus, our transcriptome expression analysis of 2D and 3D cultures, in addition to previous expression microarray data on the 21EM15 cell line [23] as well as the established knowledge on the function of several crystallins [51,52,53,54,55,56,57,58,59] offer an explanation toward the basis of refractive properties in these organoids. More importantly, these results also indicate that a subset of genes associated with lens development and fiber cell differentiation are induced under 3D culture conditions (Figure 2D). Among this set of genes, we found *Six3*, a well-characterized lens development gene [60]; *Cp*, which is typically expressed in lens epithelium [61]; *Cryab*, *Crim1* and *Nupr1*, which are expressed in lens fiber cells [26,32,61,62] and *Dkk3*, which is a component of the Wnt signaling pathway involved in lens development [62] (Figure 2B). An RNA-seq analysis also shows that the genes elevated in the external region of the lens organoids are enriched for candidates that are related to the cell cycle function (*Ccna2*, *Ccnb1*, *Ccnb2*, *Ccnd1*, *Ccne2*, *Cdc20* and *Cdk1*), whereas those elevated in the internal region are enriched for candidates involved in the Pax6 regulatory pathway (*Cap2* and *Meis1*), Notch and Wnt signaling (*Jag1*, *Wls* and *Fzd6*), Maf pathway (*Maf* and *Eya1*), lens fiber cell morphology and physiology (*Ank2*, *Atp1b1*, *Prox1* and *Tdrd7*) and organelle degradation (*Hsf4* and *Fundc1*) (Figure 3F and Appendix A). These results are relevant to lens organization, as the anterior epithelium is involved in lens growth and the cortex is the place where fiber cells progressively differentiate [3,4].

Accordingly, KI67 staining shows that only the external cells are proliferative (Figure 3D). We also found the outermost region to be positive for Laminin staining (Figure 4A). Laminin is a glycoprotein specific of basal lamina. It is present in the capsule, a cell-free structure around the lens [35]. In 21EM15 organoids, Laminin appears to be secreted by cells in the first two or three layers that are encompassed into this extracellular matrix. We were unable to detect E-cadherin or FOXE3 labeling in these external-most cell layers, suggesting that organoid epithelial cells do not fully recapitulate properties of lens anterior epithelium, which express E-cadherin or FOXE3 [63,64] (data not shown). This suggests that while 21EM15 cells are able to proliferate or enter the early stages of fiber cell differentiation, they cannot become true epithelial cells in the culture conditions that we used. One possible explanation relates to the fact that 21EM15 cells were likely selected based on their ability to proliferate rapidly [23]. Although it is now established that the anterior epithelium contains stem cells [2,22], the 21EM15 cells probably originate from the germinative zone and are therefore engaged in the early stages of the fiber cell differentiation process, preventing them from committing to the epithelial fate. Interestingly, they express *Pax6* and *Bmi1*, which are required for lens epithelial cells to regenerate a functional lens [22], but also typical stem cell marker genes like *Nes*, *Chrd*, *Sox4, Sox9, Sox12* and *Klf4* [23]. They are also able to spontaneously form lentoid bodies [23,24]. Finally, we show here, based on a histological analysis, that symmetry is broken in 3D 21EM15 cell cultures (Figure 3A, Figure 4 and Figure 5). Symmetry breaking is a general hallmark of organoid development [31,32,33]. Taken together, these observations suggest that 21EM15 cells possess stem cell-like properties accounting for their ability to form lens organoids.

A key property of lens fiber cells is the elimination of organelles, as they are potential sources of light scattering. We tested if this also applies to the core region of 21EM15 organoids. Several gene regulatory networks involved in autophagy or nuclear degradation in the lens have been identified [38,65]. One of them involves the transcription factor HSF4, a major regulator of membrane organelle degradation in the lens [39,43,66,67]. Accordingly, we found in immunostaining experiments that HSF4 is more abundant in the core than in the peripheral region of the organoid (Figure 5E). At the RNA level, *Bnip3l*, *Lamp1*, *Fundc1* and *Smurf1*, which are also involved in organelle degradation [38,65], also exhibit elevated expression in the core region compared to the outer region of organoids (Appendix A). The high expression of these genes is accompanied by an apparently complete degradation of the mitochondria, as inferred from the loss of TOMM20 staining, which is a mitochondrial marker that is commonly used to assess mitophagy in various cell types including lens fiber cells [38,41,68] (Figure 5D). The shape of the nuclei is also strongly affected in the organoid core, with some nuclei clearly showing a pyknosis-like appearance (Figure 5B). Pyknotic nuclei are characteristic of various types of terminal cell differentiation processes requiring nucleus degradation, as in red cells or lens fiber cells [69]. In addition to organelle degradation, fiber cell differentiation is also featured by a strong expression of crystallins, which involves PAX6 and, to some extent, HSF4 [34]. αB-Crystallin (*Cryab*) is poorly expressed in the outer region, whereas it is enriched in the central region where *Pax6* is expressed, as revealed by both 3′ end RNAseq on laser-microdissected regions (Supplemental Appendix A; FDR = 0.03) and IF (Figure 4D). This is similar to the expression pattern of αb-Crystallin in wild-type lenses, wherein it is high in fiber cells compared to epithelial cells in later stages of development [30]. Together, these data show that, while the outermost layers of 21EM15 organoids resemble lens epithelial cells, their core regions resemble fiber cells.

In the lens, the transition zone is the area where cells of the anterior epithelium exit the cell cycle and begin differentiation into fiber cells that contribute to the bulk of the lens tissue. Beyond the transition zone, the cells sequentially elongate and degrade their organelles while expressing key lens development master genes. We were unable to profile the transcriptome of the intermediate region of 21EM15 organoids due to technical limitations in laser microdissection. However, we gained significant insights into this zone via the immunostaining of PAX6, JAG1 and PROX1 (Figure 4). These key genes function to orchestrate the development of these different regions of the lens [1,70]. The function of JAG1 is to keep epithelial cells from the germinative zone undifferentiated and to make sure they keep proliferating by activating Notch signaling [66,67,70]. PAX6 and PROX1, respectively, trigger cell cycle exit and are associated with the expression of crystallins and cell elongation during secondary fiber cell differentiation [68,71,72,73,74]. PROX1 exhibits dynamic expression and localization in lens cells. PROX1 is first located in the cytoplasm in the anterior epithelium, and then becomes nuclear in the transition zone, where it is involved in orchestrating fiber cell elongation and differentiation [36,73]. 21EM15 organoids show an interesting distribution of these three key genes. JAG1 is consistently localized in the lateral regions and severely reduced near the central axis, in contrast to PAX6, which is highly expressed in a ring-like manner surrounding the organoid core and in the central axis. The changes in PROX1 localization seem to recapitulate its endogenous lens expression. While there is diffused cytoplasmic labeling of PROX1 in the *Jag1*-expressing region, PROX1 becomes nuclear along the central axis, a region where *Pax6* is quite highly expressed. It is interesting to note that the cells that exhibit nuclear PROX1 are located in the area where the cells are the most elongated, indicating that the organoids recapitulate this functional aspect of PROX1, similar to endogenous lens development (Figure 3A right and Figure 4B).

From a developmental point of view, 21EM15 organoids do not form the lens vesicle that is typical of mammals, and their development is more reminiscent of what happens in the fish eye [75]. This is probably due to the fact that the environment of the organoid is very simplified compared with what happens in the whole eye. Nevertheless, the fact that differentiation events occur, and that the expression of key lens genes is spatially regulated, accompanied by significant cellular changes (nuclei and mitochondria), makes this system a valuable tool for more in-depth studies. In particular, it could be used to understand the signaling pathways responsible for the cell polarity of the lens, its morphological asymmetry and the acquisition of its optical properties.

Overall, these results show that 21EM15 organoids recapitulate several aspects of lens organization, gene expression profiles, biological processes or optical properties. We therefore summarized these data in a model (Figure 7). In this model, we were unable to formally identify a canonical epithelium. We termed the outermost zone comprising the first layers of cells that are proliferative and are embedded in Laminin as the “capsuloid”. The capsuloid likely corresponds to the fusion of the capsule and the germinative zone of the lens epithelium. The lateral region expressing *Jag1* and *Ki67* is probably a mix of more or less proliferating cells (as indicated by their expression of *Ki67*) and early differentiating cells (expressing *Jag1*). Although cells from the germinative zone do not express *Jag1*, it is tempting to compare this region of the organoid to the germinative zone in the lens, where cells that express *Jag1* would prevent cell cycle exit and the fiber cell differentiation of proliferating cells. In a more internal region, between the germinative zone and the central axis of the organoid, cells begin to express *Pax6* and *Cryab*, and PROX1 becomes nuclear. This region is characteristic of the transition zone in endogenous lens development, where the cells progressively engage in the process of fiber cell differentiation. Interestingly, along the central axis and around the organoid nucleus, the cells no longer express *Jag1*. However, these cells still express *Pax6*, PROX1 is nuclear, and most importantly, exhibits elongation (Figure 4B). Cells in the central-most region exhibit a very intensely stained “pink” cytoplasm in histological analysis, reminiscent of the lens, with pyknotic nuclei and strong expressions of *Hsf4* and *Cryab*. They also show features that are consistent with the degradation of their organelles (loss of Lamin B, TOMM20 and Fibrillarin markers). All of these data suggest ongoing cellular and molecular processes in the lens organoids that contribute to transparency.

Finally, we addressed the suitability of 21EM15 organoids as a tool for studying lens opacity or cataracts (Figure 6). We observed that two previously established treatments to induce cataract, namely, the exposure to H_2_O_2_ and NaCl, are also able to induce opacity in 21EM15 organoids. This is significant, as previously, H_2_O_2_ was shown to model the events leading to age-related cataract [76,77,78]. Further, treatment with hyperosmotic NaCl was shown to trigger osmotic stress and disrupt the fluid balance of lens, which is frequently associated with dry eye disease and diabetic cataracts [79,80,81]. Thus, 21EM15 organoids could be applied to further understand how these processes induce cataract. Finally, we also observed that the knockdown of the cataract-linked RNA-binding protein encoding gene, *Celf1*, reduces 21EM15 organoids’ transparency and light-focusing properties. CELF1 is involved in the regulation of mRNA splicing, stability or translation [82], and its inactivation in mice leads to the misregulation of post-transcriptional gene expression control in the lens and cataract [46,47,83]. These results indicate that 21EM15 organoids respond to cataractogenic conditions that are representative of a wide range of lens-related etiologies (e.g., age-related cataract, diabetic cataract and genetic cataract) and thus can be used to further advance knowledge on lens pathology.

## 5. Conclusions

In conclusion, our work presents a new mouse organoid model that is effective and easy to set up and does not require the development of technical skills in stem cell culture. For a limited investment, both in terms of technique and time, this allows us to relatively rapidly obtain lens organoids that recapitulate specific aspects of lens biology, with the added possibility of performing a functional genetic analysis in a cost-effective manner. The 21EM15 organoids therefore provide the lens community with a compelling new model to improve the understanding of lens biology. Cataract remains a major public health problem that is currently only treated with surgery. It would therefore be interesting to develop drug treatments, particularly in order to offer alternatives to populations in countries that do not have ready access to surgery, or to prevent cataract formation in populations exposed to cataractogenic conditions. From a clinical point of view, these lens organoids should make it possible to develop screens to identify compounds that impact the prevention and/or treatment of cataracts.

## Figures and Tables

**Figure 1 cells-12-02478-f001:**
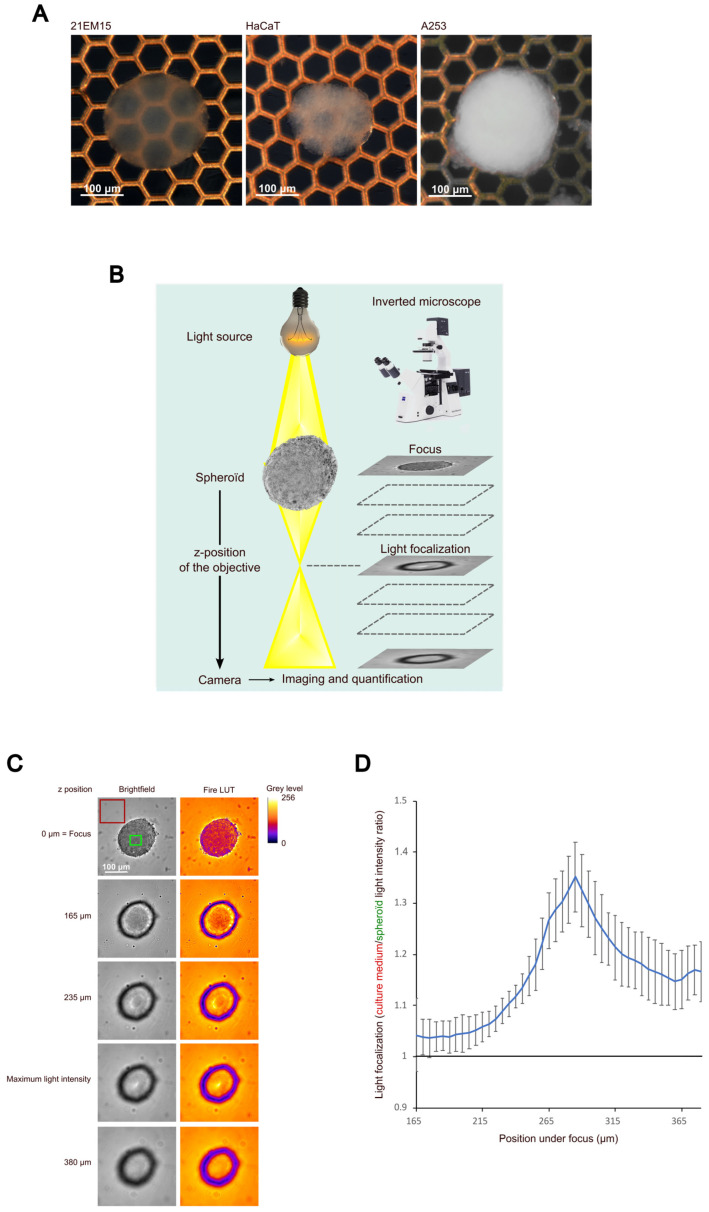
21EM15 spheroids are transparent and can focus light. (**A**) Macroscopic views of 10-day old 21EM15, HaCaT or A253 spheroids formed with 10,000 cells. The electron microscopy grid allows for the evaluation of transparency. (**B**) Schematic of the imaging setup for quantifying the light-focusing ability of spheroids. (**C**) Microscopic images of a 21EM15 spheroid showing its ability to transmit and focus light. The mean intensity of light transmitted by the medium, and the maximum intensity of light transmitted by the spheroid used for quantification are, respectively, indicated by the red and the green squares. (**D**) Graph showing the light-focusing ability of the 21EM15 spheroids calculated as the ratio between the mean intensity measured in the red square and the maximum intensity measured in the green square. This graph is representative of 5 independent experiments with *n* = 12 spheroids for each experiment. Error bars represent standard deviations. All spheroids were generated from 10,000 cells.

**Figure 2 cells-12-02478-f002:**
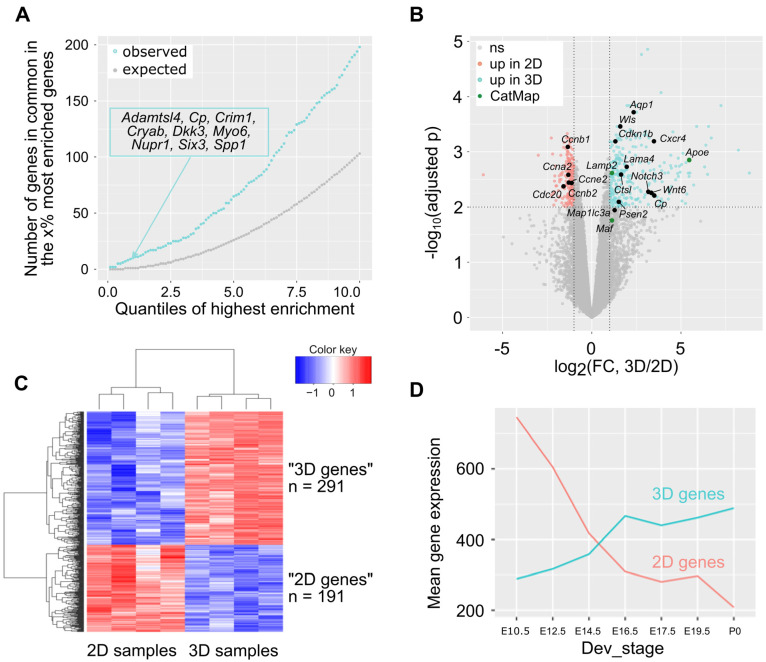
Growing 21EM15 cells in 3D culture conditions captures aspects of gene expression in lens development. (**A**) We ranked all the genes expressed in mouse E14.5 lens and in 21EM15 3D cultures (*n* = 10,320) based on lens enrichment (expression in the lens compared to whole body) as described in [28]. We separately listed the percentages (x%) of the most enriched genes in the lens and in 3D cultures for several x values ranging from 0 to 10 (X-axis). For each x value, we retrieved the genes in common between the x% of most enriched genes in lens and the x% of most enriched genes in 3D cultures. The Y-axis shows the number of shared genes for each x value. Blue represents the observed number. Grey represents the number expected if enrichment in 3D 21EM15 cultures is independent from enrichment in E14.5 lenses. The inset is the list of genes that are in both the 1% most enriched genes in 3D 21EM15 cultures and in the 1% most enriched genes in E14.5 lenses. (**B**) Volcano plot showing the statistical significance (Benjamini–Hochberg-adjusted *p*-value; −log10 scale) against the fold change (log2 scale) of the expression in 2D and 3D samples. This analysis identifies 482 differentially expressed genes (FDR < 0.01 and log2(FC) > 1 in absolute value), among which 291 are upregulated in 3D samples and 191 are upregulated in 2D samples. Genes that are found to be linked to cataract in the Cat-Map database are indicated. (**C**) Heat map of differentially expressed genes in 21EM15 spheroids (3D samples) and 21EM15 2D cultures. (**D**) Mean expression of the 191 “2D genes” and the 291 “3D genes” throughout lens development. (**A**,**D**) data for gene expression in lens are from iSyTE 2.0 [28].

**Figure 3 cells-12-02478-f003:**
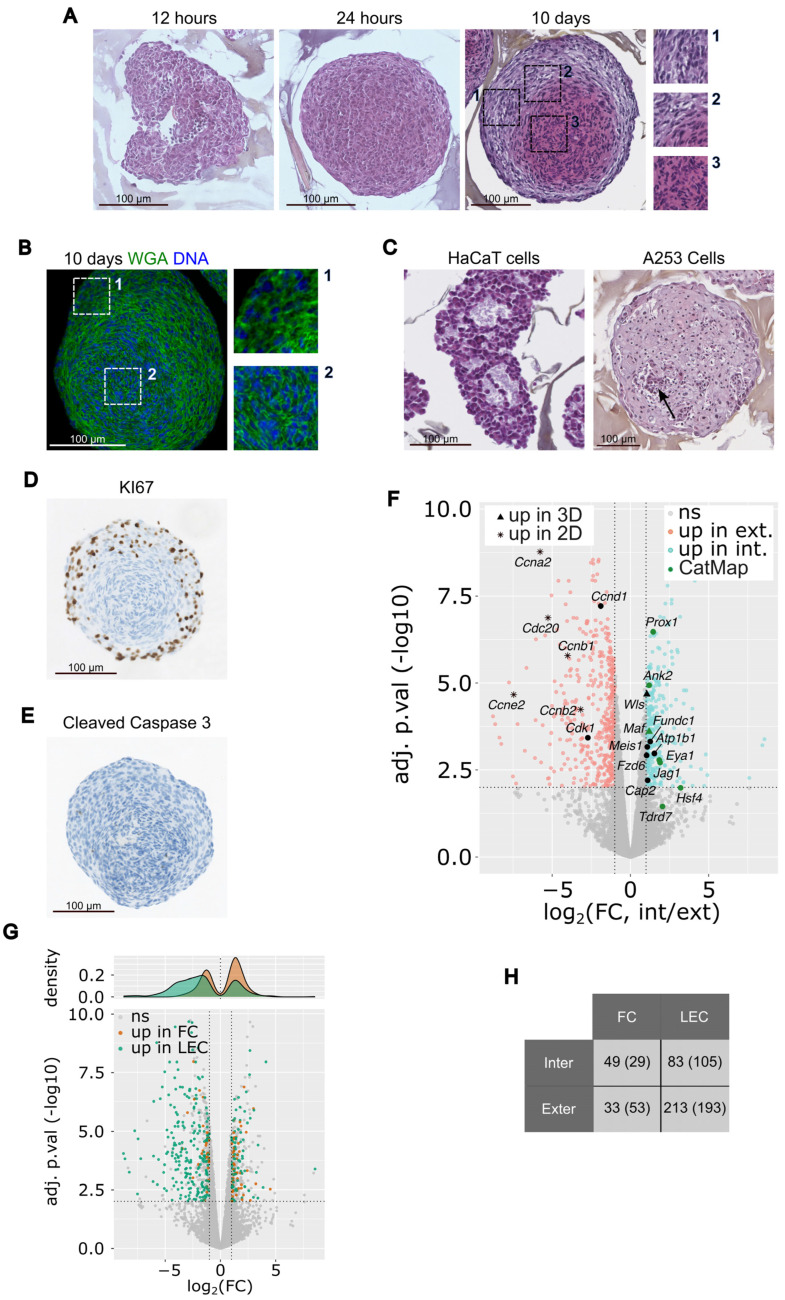
21EM15 LECs 3D cultures differentiate to form multilayered lens organoids. (**A**) Histological analysis of 21EM15 spheroids grown for 12 h, 24 h or 10 days, stained with Hematoxylin, Eosin and Safran (HES). The insets show the three different histological regions. (**B**) Microscopic image of a 21EM15 spheroid stained with Wheat Germ Agglutinin (WGA) to show distinct cellular boundaries. Insets show details of the outer (1) or inner regions (2). (**C**) Histological analysis of HaCaT or A253 spheroids grown for 10 days (HES staining). The arrow shows a necrotic region. (**D**) Histological analysis showing the localization of KI67 in a 10-day-old 21EM15 spheroid. (**E**) Cleaved Caspase 3 staining of 10-day-old 21EM15 spheroid, revealing an absence of cells undergoing apoptosis. For (**A**–**E**), data are representative of at least 3 independent experiments; *n* = 30 organoids. (**F**) Volcano plot showing the statistical significance (Benjamini–Hochberg-adjusted *p*-value; −log10 scale) against the fold change (FC; log2 scale) of the expression in internal and external regions microdissected from 21EM15 spheroids. This analysis identifies 793 differentially expressed genes (FDR < 0.01 and log2(FC) > 1 in absolute values), among which 328 are upregulated in internal regions and 465 are upregulated in external regions. “Up in 2D” and “Up in 3D” correspond to genes, respectively, upregulated in 2D culture or upregulated in 3D culture in Figure 2B. Black solid symbols correspond to genes that are of particular significance regarding the lens. (**G**) Lower panel, same volcano plot as in F, with genes that are overexpressed in microdissected lens epithelial cells (LEC) and fiber cells (FC) [30] colored in green and orange, respectively. Higher panel, density plot showing the distribution of fold changes (log2 scale) for FC and LEC genes. (**H**) Contingency table showing the number of genes that are enriched in FC or LEC compared to internal and external regions. The numbers in the brackets represent the value that is expected in the event that the enrichment in FC or LEC is unrelated to the enrichment in internal or external regions, respectively. The difference between the expected and the observed value indicates that a higher number of genes than expected are found to be enriched both in FC and internal regions, as well as in LEC and external regions.

**Figure 4 cells-12-02478-f004:**
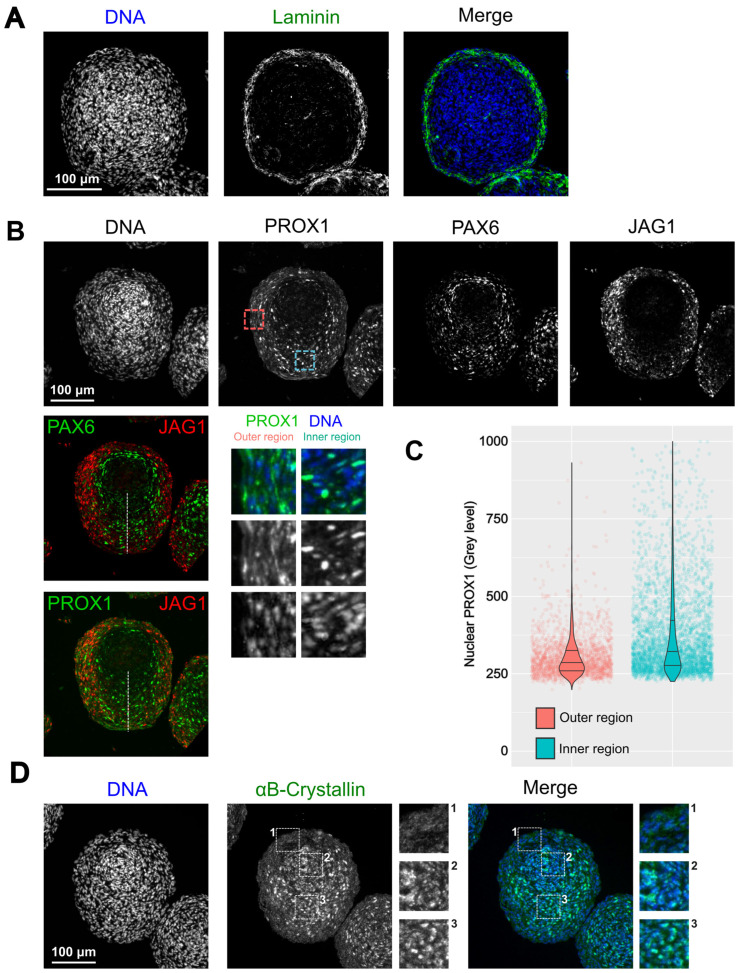
Morphological organization of 21EM15 organoids partially recapitulates eye lens patterning. (**A**) Immunofluorescence (IF) microscopy reveals the localization of Laminin on the outer region of the organoid. (**B**) Multiplex microscopic images indicate the localization of the expressed proteins, PAX6, PROX1 and JAG1. The red and blue squares correspond to the typical areas (respectively, the outer and inner regions) used to display the magnification insets presented below the grey level image of PROX1 and to quantify the PROX1 nuclear labeling shown in (**C**). The white dashed line symbolizes the central axis of the organoid. (**C**) Violin plots combined with jittered scattered plots showing that PROX1 is enriched in the nuclei of cells located in the inner region. This graph is representative of three independent experiments; *n* = 30 organoids. *p* < 2.2 × 10^−16^; Wilcoxon rank sum test with continuity correction. (**D**) IF images showing the localization of αB-Crystallin. Insets show the enlargement of the outer region, the central axis and the core of the organoid. Data are representative of at least three independent experiments; *n* = 30 organoids.

**Figure 5 cells-12-02478-f005:**
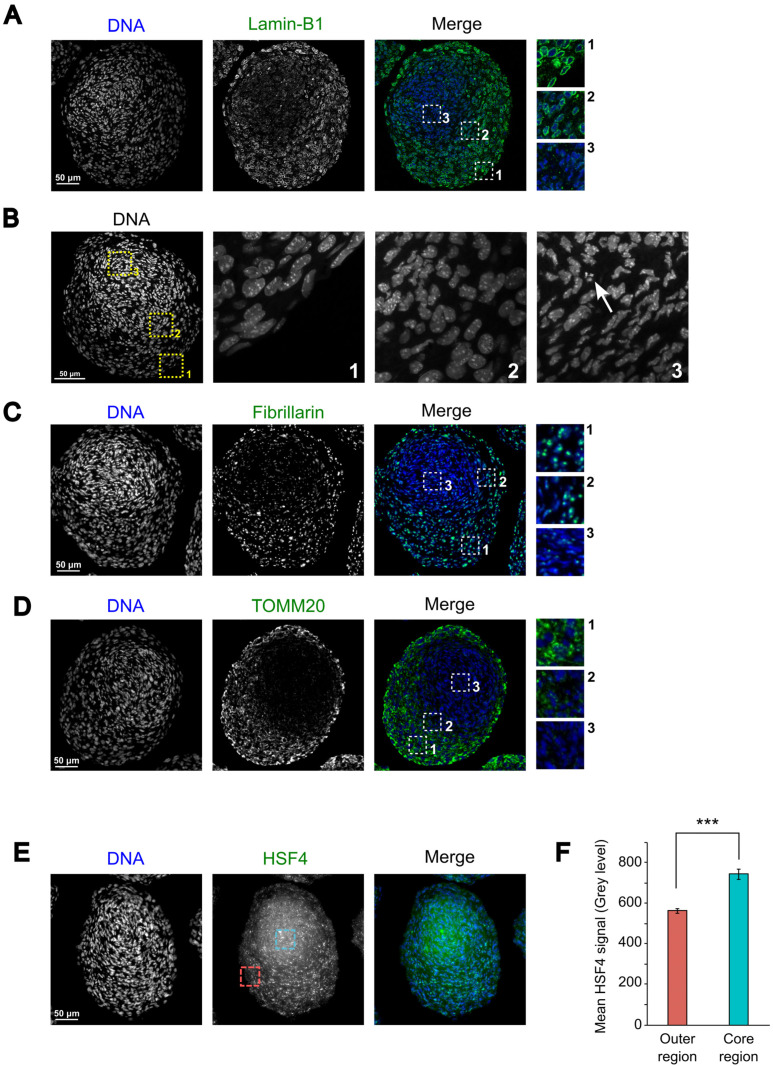
Cells located in the core of lens organoid are engaged in a process of organelle degradation. (**A**) IF microscopy images of 10-day-old lens organoid showing the localization of Lamin-B1. (**B**) Microscopy images showing the appearance of nuclei in three different areas of an organoid. The arrow points out a nucleus with pyknotic features. (**C**) Nuclei located in the core of the organoid show reduced transcriptional activity, as evidenced by decreased levels of Fibrillarin, which has previously been used as a marker of the transcriptional state of lens fiber cells. (**D**) The decrease in TOMM20 labeling indicates that the organoid core cells are engaged in mitochondrial degradation. (**E**) IF microscopy images showing the localization of HSF4. The red and blue squares correspond to the typical areas (respectively, the outer and inner regions) used to quantify the HSF4 mean signal quantified in (**F**). (**F**) Histogram presenting the quantification of the mean HSF4 signal in the outer and core regions of the organoids. This graph is representative of three independent experiments; *n* = 30 organoids. Asterisks indicate a *p*-value of <0.001. Error bars represent standard deviations.

**Figure 6 cells-12-02478-f006:**
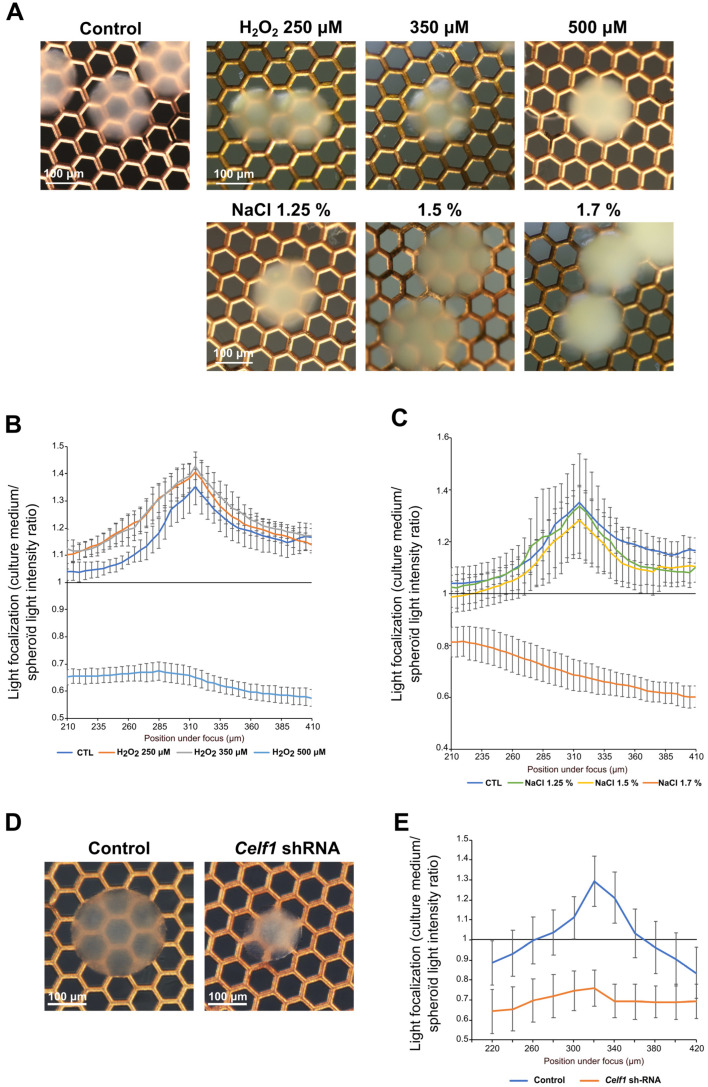
21EM15 lens organoids can be induced to develop opacity. (**A**) Macroscopic views of 10-day-old lens organoids treated with the indicated concentrations of H_2_O_2_ or NaCl. The electron microscopy grid allows for the evaluation of transparency. (**B**) Graph showing the light-focusing ability of the 21EM15 lens organoids treated with increasing concentrations of H_2_O_2_. (**C**) Graph showing the light-focusing ability of the 21EM15 lens organoids treated with increasing concentrations of NaCl. (**D**) Macroscopic views of lens organoids expressing control or *Celf1*-targeting shRNA. The electron microscopy grid allows for the evaluation of transparency. (**E**) Graph showing the light-focusing ability of the 21EM15 lens organoids expressing control or *Celf1*-targeting shRNA. Graphs B, C and E are representative of three independent experiments with *n* = 12 organoids that are 10 days old for each experiment. Error bars represent standard deviations.

**Figure 7 cells-12-02478-f007:**
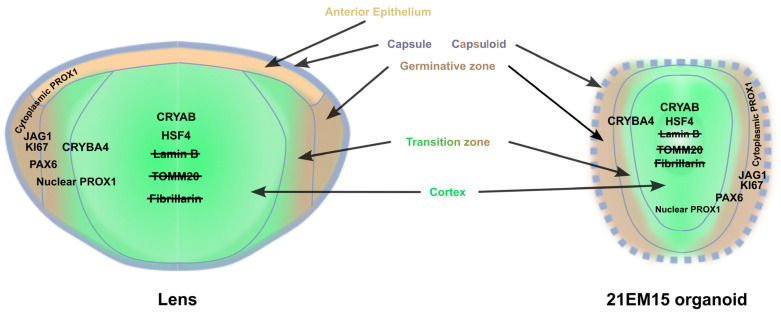
Model for 21EM15 lens organoid organization. This model shows that, unlike the lens, 21EM15 organoids lack the typical lens capsule and anterior epithelium. Instead, there is a zone that we term “capsuloid”, comprising the outer layers of proliferative cells embedded in a Laminin layer. Further inwards and from the outside in, 21EM15 organoids recapitulate several aspects of the organization of the lens, with a region corresponding to the germinal zone, where *Ki67* and *Jag1* are expressed, followed by a transition zone that expresses *Pax6* and where PROX1 progressively becomes nuclear. Along the central axis of the organoid, PROX1 is predominantly nuclear, and cells progressively degrade their organelles, such as nuclei (loss of Lamin B and Fibrillarin) and mitochondria (loss of TOMM20 signal), and begin to express *Hsf4* and *Cryab*. These events correspond closely to those described in the normal lens cortex.

## Data Availability

The RNA-seq data are available on the NCBI Gene Expression Omnibus (GEO) database under GSE228547 series.

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
