# Peer review of "Eye Lens Organoids Made Simple: Characterization of a New Three-Dimensional Organoid Model for Lens Development and Pathology"

_cells, 2023, doi:10.3390/cells12202478_

Round 1
Reviewer 1 Report
The submission describes a system to generate mouse lens cell spheroids that have some optical properties lending them potentially too high-throughout screening assays for drugs that could either improve the optical properties as well as identifying those chemicals/drugs that induced the loss of transparency and optical properties. The submission demonstrates its utility in identifying important genes and their networks in lens properties.
The data are presented logically and the narrative carefully planned and presented. There's however one very major and significant issue with their experimental system and that is the use of copper EM grids to image the spheroids given that the mouse lens system is notoriously sensitive to exposure to metal instruments and subsequent loss of transparency. So for me this specific technical issue needs to be addressed. If I have made a wrong assumption regarding the EM grids being copper, then please accept my apologies, but this detail, including the mesh size, were omitted from the manuscript. The refractive index for the spheroids would also be a useful parameter to measure as it is clear from the histological details included that the cell organisation is somewhat different to a mouse lens in situ.
The comparison of the histology for the mouse lens cells, HaCaT and A253 cells is very important (Fig 3). The images in Fig. 3A do not allow a comment regarding whether the cellular aspect ratio might change across the spheroid, but it is clear that there are nuclear shape and mitochondrial changes as well as cell compaction. The epithelium of a mouse lens covers the anterior hemisphere and the epithelial cells at the lens equator give rise to the lens fibre cells, which are highly elongated and an iconic characteristic of lens cell differentiation, even before there are changes to the nuclei and mitochondria. The epithelial cell distribution and elongated fibre cell features are not apparent in the spheroids/organoids as recognised in the Discussion and the most logical conclusion is that the absence of the spatial and chemical cues afforded by the positioning of the lens in a living mouse eye, the influence of the humors, the retina and the ciliary body all contribute to explain these important differences. Therefore the refractive index of the organoids is an important measurement to include as evidence of the optical properties arising from the organdie organisation the authors purport to show in this submission.
This latter point is exceedingly important to the submission and the value of this model. The system is atypical of mammalian lens development as a lens vesicle is not formed, rather a spheroid in the first 24 hours, resonating to a degree with teleost lens development. The fact that differentiation proceeds and key lens genes appear to be spatial regulated in the cell mass with important cell biological changes to nuclei and mitochondria makes this a valuable system for further, more detailed investigation. Because of this, I lean more to thinking that the system will deliver greater understanding of the genes and pathways responsible for lens cellular polarity and morphological asymmetry rather than understanding cataractogenesis per se. It should also lead to deeper understanding of lens transparency and development of its optical properties, especially as the Supplementary date in tables 1 and 2 suggest that gamma-crystallins are not expressed as is the case during mouse lens development rather there are beta and zeta crystallins. Surely this deserves some discussion. Perhaps lens formation, its transparency and optical properties are more plastic than we have previously conceived, so this is a valuable and very important experimental tool, adding to those already described.
Others have developed similar experimental systems and I think it would be wise to include a more detailed comparison in the Discussion. It is clear that 3D culture methods are an important feature for future models and their investigation. To this end, Figure 7 is helpful, and perhaps embellishing this figure to include other transcriptional information, such as the expression of members of the CRYB (CRYBG1/G3 and A4) CRYL (CRYL1), and CRYZ (CRYZ and CRYZL2) as well as indicating the absence of CRYG protein families would be helpful, especially when combined with any refractive index measurements that are made.
There are some other points that need to be addressed.
In the Introduction, on page 2 line 63/64, the concluding sentence is inaccurate as that was the very point of the Murphy et al study, evidenced by subsequent publications. Importantly that system is HUMAN, not mouse and therefore it is as meritorious as the current study using the 21EM15 cells. Indeed some would argue that the spontaneous differentiation and immortalisation capability of mouse lens epithelial cells, means the human cell system is the more valuable negating any technical differences. So it might be wise to modify this sentence. To conclude on lines 84/85 that "these organoids can be potentially used for screening compounds that could have an effect on both the prevention and/or treatment of cataract." should be qualified as it is perhaps unwise to assume that mouse and human lens cell systems are exactly the same.
Murphy et al. A simplified method for producing human lens epithelial cells and light-focusing micro-lenses from pluripotent stem cells. Exp Eye Res. 2021 Jan;202:108317. doi: 10.1016/j.exer.2020.108317. Epub 2020 Oct 29. PMID: 33130031.
Shparberg R, Dewi CU, Gnanasambandapillai V, Liyanage L, O'Connor MD. Single cell RNA-sequencing data generated from human pluripotent stem cell-derived lens epithelial cells. Data Brief. 2021 Jan 8;34:106657. doi: 10.1016/j.dib.2020.106657. PMID: 33521174; PMCID: PMC7820909.
In the Materials and Methods, the optical properties of the lens spheroids were measured using a AZ100 Nikon microscope. The light source is uncollimated, but more importantly, the copper grid could potentially limit the field of view and therefore affect the measurements for different micro-lenses depending on their size and positioning on the EM grid itself. The authors might like to consider citing the recent paper from Young et al used to measure the optical properties of isolated vertebrate lenses in subsequent technical updates. This is because from experience, freshly isolated mouse lenses are peculiarly sensitive to opacification when handled with metal instruments. Copper is a known cataract-causing metal there is the suggestion from the images presented that this might also be true for the spheroids of mouse lens cells (cf Figs 1A and 6A). This does give significant reason for concern as expressed above.
Young LK, Jarrin M, Saunter CD, Quinlan RA, Girkin JM. Non-invasive in vivo quantification of the developing optical properties and graded index of the embryonic eye lens using SPIM. Biomedical Optics Express 2018; 9: 2176-2188
Palomino-Vizcaino G, Schuth N, Domínguez-Calva JA, Rodríguez-Meza O, Martínez-Jurado E, Serebryany E, King JA, Kroll T, Costas M, Quintanar L. Copper Reductase Activity and Free Radical Chemistry by Cataract-Associated Human Lens γ-Crystallins. J Am Chem Soc. 2023 Mar 29;145(12):6781-6797. doi: 10.1021/jacs.2c13397. Epub 2023 Mar 14. PMID: 36918380.
Ramirez-Bello V, Martinez-Seoane J, Fernández-Silva A, Amero C. Zinc and Copper Ions Induce Aggregation of Human β-Crystallins. Molecules. 2022 May 6;27(9):2970. doi: 10.3390/molecules27092970. PMID: 35566320; PMCID: PMC9105653.
Author Response
Thank you very much for taking the time to review this manuscript. Please find the detailed responses below and the corresponding revisions/corrections shown in red in the re-submitted file.
Reviewer 1
The submission describes a system to generate mouse lens cell spheroids that have some optical properties lending them potentially too high-throughout screening assays for drugs that could either improve the optical properties as well as identifying those chemicals/drugs that induced the loss of transparency and optical properties. The submission demonstrates its utility in identifying important genes and their networks in lens properties.
The data are presented logically and the narrative carefully planned and presented. There's however one very major and significant issue with their experimental system and that is the use of copper EM grids to image the spheroids given that the mouse lens system is notoriously sensitive to exposure to metal instruments and subsequent loss of transparency. So for me this specific technical issue needs to be addressed. If I have made a wrong assumption regarding the EM grids being copper, then please accept my apologies, but this detail, including the mesh size, were omitted from the manuscript. The refractive index for the spheroids would also be a useful parameter to measure as it is clear from the histological details included that the cell organisation is somewhat different to a mouse lens in situ.
We agree with the reviewer that the use of copper grids could be problematic if organoids were cultured for a long time on this type of grid. Indeed, according to the paper from Ramirez-Bello and colleagues (cited by the reviewer), copper could trigger crystallins aggregation leading to opacification of the lens. Figure 2 of this article shows that turbidity appears after several minutes of contact between copper and recombinant crystallins. In our conditions, organoids were grown as described in the methods section (not exposed to copper grid) and were only transferred onto copper grids for a maximum time period of 30 seconds for imaging. Under these conditions, the wild-type organoids remained transparent, demonstrating that there was no copper-induced toxicity during this period. The protocol has been further detailed in the methods section and the mesh size of the grids has been included (lines 160 to 162). For refraction index measurement, please see below.
The comparison of the histology for the mouse lens cells, HaCaT and A253 cells is very important (Fig 3). The images in Fig. 3A do not allow a comment regarding whether the cellular aspect ratio might change across the spheroid, but it is clear that there are nuclear shape and mitochondrial changes as well as cell compaction. The epithelium of a mouse lens covers the anterior hemisphere and the epithelial cells at the lens equator give rise to the lens fibre cells, which are highly elongated and an iconic characteristic of lens cell differentiation, even before there are changes to the nuclei and mitochondria. The epithelial cell distribution and elongated fibre cell features are not apparent in the spheroids/organoids as recognised in the Discussion and the most logical conclusion is that the absence of the spatial and chemical cues afforded by the positioning of the lens in a living mouse eye, the influence of the humors, the retina and the ciliary body all contribute to explain these important differences. Therefore the refractive index of the organoids is an important measurement to include as evidence of the optical properties arising from the organdie organisation the authors purport to show in this submission.
We agree with the reviewer that 21EM15 organoids do not have a classical anterior epithelium as described for the lens. We have specifically addressed this issue in the discussion section (lines 564 to 567 and 643 to 647). We fully agree with the reviewer that refractive index measurement would be of great value for the further characterization of 21EM15 organoids, particularly with respect to their histological organization which does not exactly recapitulate the structure of the lens. However, setting up correct conditions to measure the refractive index would require a considerable amount of time and effort. Even if we consider undertaking these experiments in the future, we believe that, for the current manuscript, demonstrating that the organoids are transparent and can focus light is already a significant step towards the characterization of their optical properties. Notably, this shows that the basic structure of the organoids is sufficient to provide relevant optical properties. We have added a paragraph in the discussion section about the fact that refractive index measurement would be of great use in the future and have inserted the paper by Young and colleagues as a reference (lines 529 to 534).
This latter point is exceedingly important to the submission and the value of this model. The system is atypical of mammalian lens development as a lens vesicle is not formed, rather a spheroid in the first 24 hours, resonating to a degree with teleost lens development. The fact that differentiation proceeds and key lens genes appear to be spatial regulated in the cell mass with important cell biological changes to nuclei and mitochondria makes this a valuable system for further, more detailed investigation. Because of this, I lean more to thinking that the system will deliver greater understanding of the genes and pathways responsible for lens cellular polarity and morphological asymmetry rather than understanding cataractogenesis per se. It should also lead to deeper understanding of lens transparency and development of its optical properties, especially as the Supplementary date in tables 1 and 2 suggest that gamma-crystallins are not expressed as is the case during mouse lens development rather there are beta and zeta crystallins. Surely this deserves some discussion. Perhaps lens formation, its transparency and optical properties are more plastic than we have previously conceived, so this is a valuable and very important experimental tool, adding to those already described.
We fully agree with the reviewer. As a consequence, we have added discussion elements concerning crystallins expression and organoid development in the discussion section (lines 538 to 544 and 632 to 640)
Others have developed similar experimental systems and I think it would be wise to include a more detailed comparison in the Discussion. It is clear that 3D culture methods are an important feature for future models and their investigation. To this end, Figure 7 is helpful, and perhaps embellishing this figure to include other transcriptional information, such as the expression of members of the CRYB (CRYBG1/G3 and A4) CRYL (CRYL1), and CRYZ (CRYZ and CRYZL2) as well as indicating the absence of CRYG protein families would be helpful, especially when combined with any refractive index measurements that are made.
We agree with the reviewer that a paragraph integrating a detailed comparison between the 21EM15 organoid model and the previously published 3D model would be interesting in the discussion section. Unfortunately, this paragraph would certainly be redundant with introductory or discussion elements that are already present in the manuscript.
Concerning the figure 7, our goal was to highlight the genes/proteins differentially expressed/present in the various regions of the organoid and to show the similarity with a true lens. As mentioned above, 2D cultured 21EM15 cells express numerous crystallins that are also expressed but not differentially in the organoid. In order not to overload the scheme, we believe that only Cryab is worth showing.
There are some other points that need to be addressed.
In the Introduction, on page 2 line 63/64, the concluding sentence is inaccurate as that was the very point of the Murphy et al study, evidenced by subsequent publications. Importantly that system is HUMAN, not mouse and therefore it is as meritorious as the current study using the 21EM15 cells. Indeed some would argue that the spontaneous differentiation and immortalisation capability of mouse lens epithelial cells, means the human cell system is the more valuable negating any technical differences. So it might be wise to modify this sentence.
The sentence has been modified according to the reviewer's comment (lines 63 to 64).
To conclude on lines 84/85 that "these organoids can be potentially used for screening compounds that could have an effect on both the prevention and/or treatment of cataract." should be qualified as it is perhaps unwise to assume that mouse and human lens cell systems are exactly the same.
The sentence has been modified according to the reviewer's comment (lines 85 to 87).
Murphy et al. A simplified method for producing human lens epithelial cells and light-focusing micro-lenses from pluripotent stem cells. Exp Eye Res. 2021 Jan;202:108317. doi: 10.1016/j.exer.2020.108317. Epub 2020 Oct 29. PMID: 33130031.
Shparberg R, Dewi CU, Gnanasambandapillai V, Liyanage L, O'Connor MD. Single cell RNA-sequencing data generated from human pluripotent stem cell-derived lens epithelial cells. Data Brief. 2021 Jan 8;34:106657. doi: 10.1016/j.dib.2020.106657. PMID: 33521174; PMCID: PMC7820909.
In the Materials and Methods, the optical properties of the lens spheroids were measured using a AZ100 Nikon microscope. The light source is uncollimated, but more importantly, the copper grid could potentially limit the field of view and therefore affect the measurements for different micro-lenses depending on their size and positioning on the EM grid itself. The authors might like to consider citing the recent paper from Young et al used to measure the optical properties of isolated vertebrate lenses in subsequent technical updates. This is because from experience, freshly isolated mouse lenses are peculiarly sensitive to opacification when handled with metal instruments. Copper is a known cataract-causing metal there is the suggestion from the images presented that this might also be true for the spheroids of mouse lens cells (cf Figs 1A and 6A). This does give significant reason for concern as expressed above.
As discussed above, the manuscript has been modified according to the reviewer's comments.
Young LK, Jarrin M, Saunter CD, Quinlan RA, Girkin JM. Non-invasive in vivo quantification of the developing optical properties and graded index of the embryonic eye lens using SPIM. Biomedical Optics Express 2018; 9: 2176-2188
Palomino-Vizcaino G, Schuth N, Domínguez-Calva JA, Rodríguez-Meza O, Martínez-Jurado E, Serebryany E, King JA, Kroll T, Costas M, Quintanar L. Copper Reductase Activity and Free Radical Chemistry by Cataract-Associated Human Lens γ-Crystallins. J Am Chem Soc. 2023 Mar 29;145(12):6781-6797. doi: 10.1021/jacs.2c13397. Epub 2023 Mar 14. PMID: 36918380.
Ramirez-Bello V, Martinez-Seoane J, Fernández-Silva A, Amero C. Zinc and Copper Ions Induce Aggregation of Human β-Crystallins. Molecules. 2022 May 6;27(9):2970. doi: 10.3390/molecules27092970. PMID: 35566320; PMCID: PMC9105653.

Reviewer 2 Report
A fine MS describing an easy large scale production of mouse lens organoids as an in vitro model for lens development. Ten thousand cells of the 21EM15 mouse lens epithelial cell line are cultured in each of 96 well-plates. Plates are pretreated with polyhema to avoid sticking of cells to walls; plates are not agitated (see below). Within a few days (up to 10 days) cells reaggregate and proliferate. Amazingly, in each well forms just one regular oval shaped spheroid, resembling much a normal lens. As controls, HaCat keratinocytes and squamous carcinoma cells of the line A253 do not form similar organized structures. With development, lens spheroids become transparent and can focus light. Extensive transcriptome data from lens spheroids are compared with those from normal lenses, revealing a high overlap between gene expressions in spheres and lenses; particular gene groups reflect developmental states of the spheres. Spheroids are internally structured, which to some extent reflects the laminar structure of lenses, as well as germinative transitions occurring during normal lens development. By H2O2 or salt treatments, the spheroids lost their transparency. Therefore the authors proposed to use their spheroid model as a high throughput system for studies on cataracts. The extensive findings of this MS are well documented by organized figures, and the results are convincing. The MS in all aspects is well written; I could find almost no typing errors (the only one was in l. 671: type …organoids provide…). One question remains and needs discussion: have the authors tried to produce their spheroids under agitation? For highly organized retinal spheroids distinctive differences were documented between culturing with/without rotation and by using different culture dishes (96 wells, 3 cm dishes, etc.; see doi: 10.1039/b806988c).
Author Response
Thank you very much for taking the time to review this manuscript. Please find the detailed responses below and the corresponding revisions/corrections shown in red in the re-submitted file.
Reviewer 2
A fine MS describing an easy large scale production of mouse lens organoids as an in vitromodel for lens development. Ten thousand cells of the 21EM15 mouse lens epithelial cell line are cultured in each of 96 well-plates. Plates are pretreated with polyhema to avoid sticking of cells to walls; plates are not agitated (see below). Within a few days (up to 10 days) cells reaggregate and proliferate. Amazingly, in each well forms just one regular oval shaped spheroid, resembling much a normal lens. As controls, HaCat keratinocytes and squamous carcinoma cells of the line A253 do not form similar organized structures. With development, lens spheroids become transparent and can focus light. Extensive transcriptome data from lens spheroids are compared with those from normal lenses, revealing a high overlap between gene expressions in spheres and lenses; particular gene groups reflect developmental states of the spheres. Spheroids are internally structured, which to some extent reflects the laminar structure of lenses, as well as germinative transitions occurring during normal lens development. By H2O2 or salt treatments, the spheroids lost their transparency. Therefore the authors proposed to use their spheroid model as a high throughput system for studies on cataracts. The extensive findings of this MS are well documented by organized figures, and the results are convincing. The MS in all aspects is well written; I could find almost no typing errors (the only one was in l. 671: type …organoids provide…). One question remains and needs discussion: have the authors tried to produce their spheroids under agitation? For highly organized retinal spheroids distinctive differences were documented between culturing with/without rotation and by using different culture dishes (96 wells, 3 cm dishes, etc.; see doi: 10.1039/b806988c).
We thank the reviewer for his positive comments regarding our manuscript. Regarding agitation, we agree with the reviewer that under specific circumstances, agitation can improve the properties of the organoids. However, producing spheroids under agitation can give less reproducible results and a lack of homogeneity in the cultures. As mentioned in the paper cited by the reviewer regarding this culture method: "One major disadvantage of the above-mentioned reaggregate models is that the growing specimens have to be cultivated under motion, that the number of growing spheres is highly variable, and analysis of individual spheres during growth is almost impossible. These obstacles become particularly hindering if one wishes to develop miniaturized, inexpensive, large-scale systems for high throughput and high reproducibility, possibly with multi-parameter analysis and/or life imaging". Because our objective was to produce large amounts of lens organoids in a highly reproducible manner, we eventually decided to use round-bottom 96-well plates coated with poly(2-hydroxyethylmethacrylate) (Polyhema). Using this method allows rapid and easy reaggregation (less than 24 hours) of all the cells into spheroids and a very good reproducibility.

Reviewer 3 Report
Peer review for Cells (#2561726): Eye lens organoids going simple: characterization of a new 3-dimensional organoid model for lens development and pathology
Concerns/Comments/Suggestions
1. The LCM and bulk RNA-seq approach is solid but it seems to me that because the “zones” are not perfectly demarcated that perhaps single cell RNASeq would be even better and give an idea as to the transcriptional information as cells transition from zone to zone. Have the authors considered this?
2. While there is active proliferation around outer layers/edge of the organoids (Ki67+ cells), the cells in the central zone of the organoids have full nuclei and do not seem to undergo denucleation like mammalian fibers. That said, the organoids seem somewhat, although not completely, transparent. Can the authors account for this? If organoids were grown/cultured for a longer duration would innermost fiber-like cells denucleate and also lose organelles? Have authors considered this?
3. I’ve never known PROX1 to be identifiable in lens epithelial cells at all, let alone cytoplasmic. Although the nuclear PROX1 looks good in “fiber-like cells” perhaps a better label to demonstrate this transition would be use of the epithelial specific factor, FoxE3 versus fiber-specific. PROX1 or cMAF? E-cadherin is also epithelial-specific and could be utilized.
4. In addition to aB-Crystallin, did the authors investigate additional crystallins? One that could be interesting the gammaS which is downregulated in expression in fibers compared to epithelia (see David Beebe’s literature).
5. While I do agree that these lens organoids could be a step forward and useful tool for drug screening and genetic manipulation, the Celf1 shRNA experiment presented in Figure 6 shows a much smaller lens organoid (roughly 1/3 in size) compared to the wild-type organoid, whereas in humans most cataracts develop later in life after normal lens growth and development for much of life. Perhaps finding a way to grow organoids to a more mature state and then trying genetic manipulation would be more translationally relevant?
Author Response
Thank you very much for taking the time to review this manuscript. Please find the detailed responses below and the corresponding revisions/corrections shown in red in the re-submitted file.
Reviewer 3
Concerns/Comments/Suggestions
- The LCM and bulk RNA-seq approach is solid but it seems to me that because the “zones” are not perfectly demarcated that perhaps single cell RNASeq would be even better and give an idea as to the transcriptional information as cells transition from zone to zone. Have the authors considered this?
The cells within the inner region are highly compacted (see Figures 3A-B). We were concerned that this could preclude efficient cell-cell dissociation, which is a prerequisite for scRNAseq. Ultimately, the inner cells would then be missing in the single cell analysis. Interestingly, a recent paper reports scRNAseq analysis of lenses, but only superficial cells were profiled (PMID:: 37057399). This is consistent with the notion that internal cells of the lens, and by extension of the organoids, probably require very specific conditions for any single cell analysis. Developing a protocol for scRNAseq analysis of lens or lens organoids will require further efforts. In addition, we would like to stress that LCM enabled us to retain relevant spatial information, which may not be the case for scRNAseq.
- While there is active proliferation around outer layers/edge of the organoids (Ki67+ cells), the cells in the central zone of the organoids have full nuclei and do not seem to undergo denucleation like mammalian fibers. That said, the organoids seem somewhat, although not completely, transparent. Can the authors account for this? If organoids were grown/cultured for a longer duration would innermost fiber-like cells denucleate and also lose organelles? Have authors considered this?
We agree with the reviewer that we could not prove that 21EM15 organoids contain denucleated cells. Nevertheless, we showed that cells, in the most central area, are engaged in a process of nuclear membrane degradation, transcription stop, nuclear compaction (pyknotic nuclei) and even fragmentation. Moreover, we also showed that central cells are also engaged in a process of mitochondria degradation. Lens transparency is not only triggered by organelles degradation but also by cell compaction, which we observed in Figure 3 A and B, reorganization of cell to cell junctions and crystallins expression. It is possible that 21EM15 organoids become relatively transparent because of a combination of all these processes, even if they are not fully completed. Concerning the duration of the culture, we let organoids grow for more than 35 days without observing any further nuclear degradation.
- I’ve never known PROX1 to be identifiable in lens epithelial cells at all, let alone cytoplasmic. Although the nuclear PROX1 looks good in “fiber-like cells” perhaps a better label to demonstrate this transition would be use of the epithelial specific factor, FoxE3 versus fiber-specific. PROX1 or cMAF? E-cadherin is also epithelial-specific and could be utilized.
We thank the reviewer for his comment. As suggested, we tried to detect E-Cadherin and FOXE3 in 21EM15 organoids. We could not detect any immunofluorescence labeling (data not shown), in accordance with the transcriptomic data that did not reveal any expression of these two genes (see supplemental table 1 and 2). This data was already present in the manuscript concerning E-Cadherin, yet we have added a comment and a reference concerning FOXE3 (lines 564 and 567). As indicated in the manuscript, "This suggests that while 21EM15 cells are able to proliferate or enter the early stages of fiber cell differentiation, they cannot become true epithelial cells in the culture conditions that we used. One possible explanation relates to the fact that 21EM15 cells were likely selected based on their ability to proliferate rapidly [24]." (lines 567 to 574). Concerning PROX1 localization, it is described in the literature as being "predominately cytoplasmic in the lens placode as well as the lens epithelium and germinative zone throughout development. However during fiber cell differentiation, Prox1 protein redistributes to cell nuclei." (Duncan et al., 2002; PMID: 11850194; reference 31 in the revised version of the manuscript).
- In addition to aB-Crystallin, did the authors investigate additional crystallins? One that could be interesting the gammaS which is downregulated in expression in fibers compared to epithelia (see David Beebe’s literature).
We agree with the reviewer that it would be interesting to get more information about crystallin genes expression. A sentence listing the various crystallins expressed in 21EM15 organoids has been inserted in the discussion section (lines 538 to 543). In this manuscript, we mainly focused on the contribution of the 3D culture conditions to the 21EM15 organoid gene expression landscape and that's why we initially only described the increase in Cryab expression between the outside and the inner regions. It is nevertheless important to get in mind that 21EM15 cells have been characterized as lens epithelial cells, and probably according to the present manuscript, as epithelial cells at least in part engaged in the early first steps of fiber cell differentiation. In their paper, Terrell and colleagues (2015; PMID: 25530357) showed that 21EM15 cells already express a significant amount of crystallin genes. In that context, it is thus not surprising to observe that 21EM15 organoids express crystallins, yet the only crystallin being differentially expressed between the outside and the core region is Cryab.
- While I do agree that these lens organoids could be a step forward and useful tool for drug screening and genetic manipulation, the Celf1shRNA experiment presented in Figure 6 shows a much smaller lens organoid (roughly 1/3 in size) compared to the wild-type organoid, whereas in humans most cataracts develop later in life after normal lens growth and development for much of life. Perhaps finding a way to grow organoids to a more mature state and then trying genetic manipulation would be more translationally relevant?
Organoids expressing Celf1 shRNA are indeed smaller than WT organoids. In this context, they appear to mimic quite closely what happens in the lens of a Celf1 knockout mouse, which has smaller lens in early postnatal life compared to the control mouse (Siddam et al., 2018; PMID: 29565969). In the mouse model, this defect is likely triggered by a defect in cell proliferation (Xia et al., 2015; PMID: 26535026). Interestingly, Celf1 shRNA-expressing organoids have fewer Ki67-expressing cells than control organoids (data not shown). We agree with the reviewer that it would be very interesting to inactivate or induce the expression of a specific gene once the organoid is normally developed. It should probably be possible to consider the use of inducible gene constructs or optogenetics to answer such questions.

Round 2
Reviewer 1 Report
I would to thank the authors for their detailed responses to my comments and my concerns about the copper grids are allayed. Nevertheless the most important issue remains. How are the refractive properties produced in these organoids?
One of the other referees raised this point by asking about gS-crystallin and I am pleased to see the inclusion of the additional sentences in the Discussion. This doesn't, however, address the question because there is only one beta-crystallin and the others are proteins with bg-crystallin domains - for example AIM1 (absent in melanoma) and the others listed are connected with diseases not normally associated with the lens. So how are the refractive properties developed? What are the protein concentrations in these cells? Is it possible that Crybg1/2 can refract light?
The authors add in their response "In their paper, Terrell and colleagues (2015; PMID: 25530357) showed that 21EM15 cells already express a significant amount of crystallin genes. In that context, it is thus not surprising to observe that 21EM15 organoids express crystallins, yet the only crystallin being differentially expressed between the outside and the core region is Cryab".
Even this result is interesting by the absence of a comment about CRYAA.
Further explanation of how these refractive properties are produced is needed. If it not possible to produce RI measurements see the Schlussler article, then perhaps a biochemical measurement and analysis of the protein concentration of extracts from the outer and inner portions of there spheroids is possible. The refractive properties are important and thus far the authors have described a system where there are refractive properties but not based upon the proteins associated with refraction in the lens.
Schlüßler R, Kim K, Nötzel M, Taubenberger A, Abuhattum S, Beck T, Müller P, Maharana S, Cojoc G, Girardo S, Hermann A, Alberti S, Guck J. Correlative all-optical quantification of mass density and mechanics of subcellular compartments with fluorescence specificity. Elife. 2022 Jan 10;11:e68490. doi: 10.7554/eLife.68490. PMID: 35001870; PMCID: PMC8816383.
Round 3
Reviewer 1 Report
Thank you for responding to my second set of comments and for the explanations provided. Thank you too for the extra paragraph in the Discussion. Unfortunately this answers only in part the issue that is of significant concern. Which crystallins are responsible for the refractive properties in these organoids? The authors are confusing mechanisms (organelle removal; chaperone protection) with crystallin expression. The Discussion needs to specifically address what lens proteins, and why, contribute to the observed refraction. To help them the authors might like to think about these questions.
Which crystallins known to contribute to refraction are expressed? This means what beta- and gamma-crystallins are expressed in this system? This does not mean the protein chaperones (alphaB-crystallin) or transcription factors (AIM1/CRYBG1) or the Very Large A-Kinase Anchor Protein (CRYBG3) that can be detected by RNA-seq.
From the reply I understand that the only beta- and gamma-crystallin that could be responsible for the refraction is CRYBA4. In the human lens, it is βA3/A1, βB1, and βB2 crystallins that are the most abundant. The beta-crystallins can form homodimers, so perhaps this beta-crystallin (CRYBA4) alone is responsible for the refraction properties observed. This point needs to be made and discussed. The fact that NO gamma-crystallins are identified needs to be overtly stated as they have been previously identified as important to lens refraction. This means the authors should clearly state that signals for CRYGA, B, C, D and E were not detected. In development the gamma-crystallins are highly expressed, but as lens development proceeds so the beta-crystallins become the most abundant water soluble b/g-crystallins in the lens cortex. This again deserves comment because the 21EM15 organoids seem to adopt the secondary fibre cell expression preference for a beta-crystallin.
Here it is important to recognise that it is the specific features of the beta/gamma-crystallins that contribute to the refractive index of the lens. These two papers explain why this is so and why other proteins that contain a b/g-domain are not important eg CRYBG1/3.
Khago, D., Bierma, J. C., Roskamp, K. W., Kozlyuk, N. & Martin, R. W. Protein refractive index increment is determined by conformation as well as composition. J. Phys. Condens. Matter 30, 435101 (2018).
Mahendiran K, Elie C, Nebel J-C, Ryan A, and Pierscionek BK. Primary sequence contribution to the optical function of the eye lens. Scientific Reports, 4:5195 (2014).
The authors declined to measure the refractive index of the 21EM15 organoids and how this might develop so they are obliged to give an explanation of how the refraction is produced by the expressed genes. This has not been done in any of the revisions and therefore for me the interpretation of the results is incomplete. Figure 7 could certainly summarise the expression pattern to make this point plain to readers where thus far there is no mention of the b/g-crystallins.
Looking again at the Terrell et al paper from 8 years ago, they specifically comment that CRYBB1 expression was down-regulated as monitored by RT-PCR. Yes CRYGS was expressed and yes there were differences between RNA-seq and microarray datasets, but now 8 years on and with Dr Lachke as the communicating author on the Terrell paper, one would have hoped for greater clarity in this submission regarding the b/g-crystallin expression.
Minor Points:
Please check that gene and protein naming conventions are correct.
For example, I believe the Terrell paper only looked at b/g-crystallin gene expression - not protein
Line 604: b-Crystallin should be B-Crystallin
